# Only human after all? a pre-registered study on gaze behavior and humanity attributions to people with facial difference

**Pauline Rasset**[1]*, **Benoît Montalan**[2], **Jessica Mange**[3]

**1** Laboratoire de Psychologie: Cognition, Comportement, Communication (LP3C), Université de Rennes 2, Rennes, France, **2** Centre de Recherche sur les Fonctionnements et Dysfonctionnements Psychologiques (CRFDP UR 7475), Université de Rouen Normandie, Rouen, France, **3** Laboratoire de Psychologie de Caen Normandie (LPCN UR 7452), Université de Caen Normandie, Caen, France

\* pauline.rasset@unicaen.fr

**Data Availability Statement:** All data have been made publicly available via the Open Science Framework (OSF) and can be accessed at this link: https://osf.io/v7eab/?view_only=

## Abstract

There is a great deal of indirect evidence suggesting that people with facial difference (FD) may be dehumanized. This research aimed to provide direct evidence of the dehumanization of people with FD based on the stigmatizing reactions they elicit. More precisely, previous findings revealed that the specific way people with FD are looked upon is related to the feelings of disgust they elicit. Since disgust fosters dehumanization, our aim was to confirm the modified pattern of visual attention towards people with FD and to determine whether it was also related to humanness perception. For that purpose, a preregistered eye-tracking study ($N = 97$) using a former experimental design extended to humanity attributions was conducted. This research replicates findings showing that the face of people with FD is explored differently in comparison with other human faces. However, the hypothesis that people with FD were given fewer humanity attributions was not supported. Therefore, the hypothesis of a "dehumanizing gaze" towards people with FD–beyond humanity-related attributions–is discussed in light of these findings.

## Introduction

People living with a facial difference (FD; also known as "facial disfigurement") are stigmatized [1], which leads them to feel dehumanized (i.e., denied a full humanness status; [2]). Indeed, they report feeling like they are not treated akin to their nonvisibly different counterparts (e.g., [3]). Relatives of people with visible FD also report witnessing their next-of-kin being treated as diminished human beings (e.g., witness statements from parents of children with birthmarks, see [4]). Although the concepts of stigma and dehumanization share some commonalities [5], to our knowledge, no previous study has directly investigated the dehumanization of people with FD seen from the perspective of their perceivers. Thus, the aim of this research work was to determine whether the stigmatization of people with FD could be related to their dehumanization.

4bd75d4a8d3045b59fa2a75c791ffa81. The design and analysis plans for the experiment were preregistered on OSF and can be accessed with this link: https://osf.io/grytk.

**Funding:** This study was supported by the Normandy Region [90 000 €, 2018; grant RIN tremplin 19E00851, 2019] to JM, and a grant from the Foundation Gueules Cassées [22 910 €, 2020] to PR. These funders did not play any role in the study design, data collection and analysis, decision to publish, or preparations of the manuscript. There was no additional external funding received for this study.

**Competing interests:** The authors have declared that no competing interests exist.

## Dehumanization of people with facial differences

Two broad types of dehumanization manifestations have been described: dehumanization as a content of social perception (e.g., the attribution of nonspecific human characteristics) and dehumanization as a form of perception (i.e., the reduction of people to their constituent parts) [6]. There are several models of dehumanization as a content of social perception with a general agreement on two basic dimensions of humanness attributions [7]. More precisely, people can be dehumanized when they are denied uniquely human (UH) characteristics (e.g., refinement; i.e., characteristics that distinguish human beings from nonhuman animals) and/or characteristics that are at the essence of human nature (HN; e.g., moral sensitivity; i.e., characteristics that distinguish human beings from inert entities) [8,9]. In parallel, people can also be dehumanized in that they are perceived in a perceptually different manner than other human beings [10].

There is—to our knowledge—no evidence of lesser attributions of humanness to people with FD. However, some indirect findings support the assumption of a dehumanized perceptual processing of people with FD. First, a research team investigating the neural responses to faces with FD found diminished responses within a brain region implicated in social cognition [11], as also evidenced for highly dehumanized people (e.g., homeless people [12]). Second, there is evidence suggesting that humanness perception is rooted in holistic face perception [10], which is impeded for faces with FD [13]. Third (and accordingly), research conducted with event-related potentials showed a reduced amplitude of the N170 –a component involved in holistic processing–when people are exposed to faces with FD in comparison to when they are exposed to faces without FD [14]. Finally, attention to the eyes is related to humanizing modes of perception [15], but for faces with FD, gaze behavior is focused on the FD and drawn away from the eyes [16]. Interestingly, this specific pattern of gaze behavior seems to play a key role in the stigmatization of people with FD.

## Gaze behavior as an indicator of the dehumanization of people with a facial difference?

Madera and Hebl [17,18] revealed that the focus on FD biases memory, which in turn fosters discrimination. More recently, Rasset and colleagues [16] showed that the interaction between the focus on the FD and the lesser attention given to the eyes was correlated with a heightened affective reaction of disgust. Concretely, they asked their participants to look at faces with and without a burn-like disfigurement on the cheek. They revealed that the more participants looked at the FD, the heightened their stigmatizing reaction of disgust was, but only when they looked little at the eyes area. Since disgust is a precursor of dehumanization [19], including the dehumanization of people with facial differences [20], it is thus possible that this stigmatizing gaze could also be a dehumanizing gaze.

Thus, the aims of this research were twofold: 1) replicate previous eye-tracking research showing a stigmatizing gaze with a focus on the FD leading to the neglect of the eyes region [16], and 2) determine if this stigmatizing gaze could be related to the dehumanization of people with FD.

## Method

For more details on the method, see the supporting information.

### Ethical considerations

Ethical approval was obtained from the local institutional review committee of the laboratory of CRFDP–University of Rouen Normandy (N˚ 2020–06-A). Every participant gave twice

(before and after the experiment) their written informed consent to participate in the study. No identifying information was recorded.

## Participants

One hundred thirteen participants volunteered to participate in the study without any financial counterpart. Twelve participants were withdrawn from the study because of calibration errors, and four were withdrawn because of disfigurement. The remaining 97 participants comprised 65 females, 30 males, and 2 nonbinary participants (M = 19.61 years; SD = 3.01).

G*Power [21] was used to calculate an a priori sample size, using a standard parameter of α = .05, a power 1 – ß = .80, for a small effect of ρ = .25 [16]. This study was preregistered at https://osf.io/grytk. Data were collected from January to September 2021. Data were consulted for research purposes once they had been fully collected.

## Procedure, material and measures

All participants were recruited directly on the campus by the experimenter and volunteered to participate. First, a Tobii X120 Eye-tracker (SR Research Ltd.) was calibrated. Each trial consisted of the following sequence: a fixation cross in the center of the screen displayed for 2 seconds, then a face displayed for 5 seconds, and then a question asking for humanness attributions.

The faces were presented either in their original condition or with an FD. Eight faces were used as stimuli: four different faces presented either in their original condition or with an FD. Participants were randomly exposed to one of two sets of four faces. In each set, participants could see two faces with FD and two faces without FD; each time, one face belonged to a woman, and the other to a man. For the two faces presenting an FD, one of them had an FD pattern on the left cheek, while the other had an FD on the right cheek. Each face was presented either in its original or in its "disfigured" condition depending on the set of faces.

Each face was presented six times in fixed random order, once for each attribute, thus bringing the total of trials to 24. Each trial lasted approximately 10 seconds. After providing sociodemographic information, they were then fully debriefed, gave a second a posteriori informed consent and were finally thanked.

The length and location of the participants' visual fixations on five separate nonoverlapping areas of interest (AoIs; i.e., eyes, nose, target cheek, other cheek, and lips) were recorded (based on [15]; see S1 Fig in S1 File). For each AoI, the *total dwell time* (i.e., the sum of all dwell times for the same AOI over a trial) and the *time to first fixation* (i.e., the time period from entering the AOI until the first fixation is made) were recorded.

*Dual Model Questionnaires* were used to measure humanness-related attributes (based on the work of Haslam et al., 2008 [22]). Based on a pilot study (see supporting information), only UH traits (α = .74) and HN traits (α = .61) were selected. Participants were asked to report "the extent to which most people seeing the person on the photograph may think that the person is" the written trait on a numeric scale of 1–7 (corresponding to: 1- 'totally disagree', 4 –'neither agree nor disagree', 7 –'totally agree').

## Statistical analysis

First, the hypothesis of an impact of disfigurement on visual attention was tested. For that purpose, a 2 (Type of face: original, with FD) X 5 (AoI: eyes, nose, target cheek, other cheek, lips) repeated-measures ANOVA was run on each eye tracking variable. Greenhouse–Geisser corrections were used to correct the analyses that violated the assumption of sphericity. Second, the assumption of the impact of the FD on humanness-related attributes was tested. For that

purpose, 2 (Type of face: original, with FD) X 2 (Humanness-related attribute: UH, HN) repeated-measures ANOVA was run. Finally, the relationships between human attributes and visual attention were explored through correlations and linear regressions. For that purpose, for all measures, the scores were computed by considering the difference mean score of the faces with disfigurement minus the original faces mean score (for a similar procedure, see [23]).

## Results

### Impact of facial differences on gaze behavior

There was a main effect of AoI on the total dwell time and an interaction effect between type of face and AoI (see Table 1 for details). As expected, simple effects tests revealed that the target cheek was fixated longer when the face presented an FD than when it was presented in its original condition (see S2 Fig in S1 File). The opposite pattern emerged for the eyes, nose and lips. A larger effect size was observed for the eyes as opposed to the nose and the lips.

There were main effects of type of face and AoI on the time to first fixation and an interaction between type of face and AoI (see Table 1 for details). As expected, simple effect tests revealed that the target cheek was stared at quicker when the face had an FD than when it was presented in its original condition (see S3 Fig in S1 File). The opposite pattern emerged for the lips and nose, with a larger effect size being observed for the lips.

### Impact of disfigurement on humanity attributions

There was neither a main effect of Type of face ($F(1, 96) = 2.61$, $p = .10$) nor a main effect of Humanness-related attribute ($F(1, 96) = 2.03$, $p = .16$). However, there was an interaction effect

**Table 1. Means (standard deviations) of eye-tracking variables for each AoI depending on the type of face evaluated in the preregistered study.** The results of repeated-measures ANOVA are also included.

| AoI | Target Cheek | Eyes | Nose | Other cheek | Lips | *All* | |
|---|---|---|---|---|---|---|---|
| **Type of face** | | | | | | | |
| **Total dwell Time.** Main effect of Type of Face: $F(1, 96) = 0.74$, $p = .39$, $\eta^2_G < .001$ | | | | | | | |
| Original | 0.18 (0.24) | 2.26 (0.83) | 0.80 (0.50) | 0.18 (0.24) | 0.33 (0.23) | 0.75 (0.47) | Main effect of AoI $F(1.53, 146.75) = 266.88$ $p < .001$ $\eta^2_G = .69$ |
| Disfigured | 0.64 (0.33) | 1.91 (0.87) | 0.71 (0.48) | 0.17 (0.18) | 0.30 (0.21) | 0.75 (0.48) | |
| *All* | 0.41 (0.29) | 2,08 (0.85) | 0.75 (0.49) | 0.17 (0.21) | 0.31 (0.22) | 0.75 (0.47) | |
| *Simple effet test* | $F = 174.28$ $p < .001$ $d = 1.34$ | $F = 26.97$ $p < .001$ $d = 0.53$ | $F = 9.68$ $p < .01$ $d = 0.32$ | $F = 0.80$ $p = .37$ $d = 0.09$ | $F = 4.05$ $p = .05$ $d = 0.20$ | | Interaction effect $F(2.00, 192.10) = 54.94$ $p < .001$ $\eta^2_G = .06$ |
| **Time to First Fixation.** Main effect of Type of Face: $F(1, 96) = 43.74$, $p < .001$, $\eta^2_G = .01$ | | | | | | | |
| Original | 4.09 (0.84) | 0.87 (0.75) | 1.20 (0.75) | 4.08 (0.76) | 3.30 (0.75) | 2.70 (0.77) | Main effect of AoI $F(3.68, 257.65) = 362.59$ $p < .001$ $\eta^2_G = .69$ |
| Disfigured | 2.58 (0.92) | 0.98 (0.86) | 1.37 (0.96) | 4.10 (0.79) | 3.54 (0.90) | 2.51 (0.89) | |
| *All* | 3.33 (0.88) | 0.92 (0.81) | 1.28 (0.86) | 4.09 (0.78) | 3.42 (0.83) | 2.61 (0.83) | |
| *Simple effet test* | $F = 203.15$ $p < .001$ $d = 1.45$ | $F = 2.96$ $p = .09$ $d = 0.17$ | $F = 5.20$ $p = .02$ $d = 0.23$ | $F = 0.02$ $p = .74$ $d = 0.03$ | $F = 10.78$ $p = .001$ $d = 0.33$ | | Interaction effect $F(3.39, 325.08) = 82.86$ $p < .001$ $\eta^2_G = .13$ |

*Note*. Values for total dwell time and time to first fixation are indicated in seconds. Contrary to total dwell time, lower scores indicate more attention for time to first fixation. Time to first fixation for the eyes and nose must be considered carefully because, for some faces, the participants were asked to focus on these areas at the onset of each image presentation.

between those two variables ($F(1, 96) = 10.72$, $p = .001$, $\eta^2_G = .01$), showing that the participants attributed more HN traits to faces with a disfigurement ($M = 4.56$, $SD = 0.96$) than to faces in their original condition ($M = 4.19$, $SD = 0.87$, $p_{Bonferroni} < .05$), whereas no such difference was observed for UH traits (faces with disfigurement: $M = 4.44$, $SD = 0.94$; in original settings: $M = 4.48$, $SD = 0.85$; $p_{Bonferroni} = .99$). This pattern of results is inconsistent with our hypothesis of a dehumanization of people with FD.

### Relationship between visual attention and humanity attributions

Neither correlation between total dwell time towards any areas of the face and the humanness-related traits–, i.e., for UH: $-.14 < r < .19$, $p > .06$; for NH: $-.11 < r < .15$, $p > .13$ –nor correlation between time to first fixation towards any areas of the face and the humanness-related traits–, i.e., for UH: $-.10 < r < .17$, $ps > .09$; for NH: $-.10 < r < .17$, $ps > .09$ –reached significance (see S1 Table in S1 File for details).

## Discussion

This research replicates previous findings showing that faces with FD engender a significantly different pattern of gaze behavior in comparison to faces without FD [16]. Put differently, the presence of FD seems to disrupt the regular way in which people gaze at human faces [24]. However, there was no evidence of lesser attributions of humanness-related traits to faces with FD, which hampers our dehumanizing gaze hypothesis. In other words, this research confirms that faces with FD are significantly differently gazed at, but there was no evidence of their dehumanization. Thus, this discussion will explore some possible explanations, in line with this lack of evidence of dehumanization along with the limitations of this study in regard to the measures, the method, and the hypotheses.

First, it is possible that the measures of dehumanization as a part of social perception cannot capture the dehumanization of people with FD, or at least the scale we have chosen. Indeed, in the past, research has shown that self-reported evaluations of people with FD are biased by social desirability [25], which may also have biased humanness attributions [26]. This would explain why our findings revealed that people with FD were perceived as having even more HN traits. Moreover, these kinds of measures of dehumanization are closely related to stereotyping [27], and there are inconstancies among studies investigating stereotypes about people with FD [28,29]. Thus, future research should also consider using measures other than those consisting of attributing characteristics (for a complete review of dehumanization-related measures, see [30]). Since people with FD are stigmatized [1] and automatic strong stigma reactions can be lowered by more controlled reactions [31], a special interest should be given to the measures allowing the investigation of dehumanization at an earlier stage of processing. More generally, measures to circumvent the social desirability bias would be advisable.

Second, we used faces displaying direct eye gaze, which may have biased humanity attributions. Faces are individuating stimuli [32]. Moreover, Khalid and colleagues (2016) [33] revealed that direct eye gaze signals an invitation to social interaction and thus promotes mind perception. Eye contact may have impeded social distancing, which would explain the unexpected overattribution of HN traits to people with FD. In other words, this direct eye gaze may have counterbalanced the negative effect of FD on mind perception. Moreover, we provided photographs of people with relatively mild forms of disfigurement, which could have been sufficient to foster a global change in gaze behavior towards people with FD but insufficient to dehumanize them. Since the more severe the disfigurement is, the more it elicits disgust [23,34], future studies should investigate the impact of a severity gradient on humanity attributions. In this research, we did not include a measure of disgust for theoretical (e.g., we were

concerned that priming our participants with disgust might bias their visual attention) and technical reasons (e.g., a longer procedure might have exhausted participants–this would have been particularly true in the case of including multiple affects). However, given that previous research on disgust [20] led us assume a dehumanization of people with FD that was not supported by our data, the impact of disgust on social perceptions of people with FD would merit future attention.

Finally, it is possible that this stigmatizing gaze towards people with FD is not underpinned by dehumanizing attributions. Thus, other related–but independent–constructs, such as objectification [6], may better correspond to the lived experience of people with FD, which deserves future attention. Regardless, people with FD feel dehumanized (i.e., metadehumanization [35]), which may lead them to perceive themselves as diminished human beings (i.e., self-dehumanization [36]). Thus, the impact of this stigmatizing gaze on their self- and meta-dehumanization should also be investigated in future studies.

## Conclusion

There is a French expression for saying that a person is unrecognizable after an event that has profoundly altered his or her facial appearance: he or she 'no longer has a human face' ('Il/Elle n'a plus figure humaine' in French). However, future research is needed to confirm that this expression is psychologically true.

## Supporting information

**S1 File. Supplementary file.** Pilot study, full method section, and supplementary results. (DOCX)

## Acknowledgments

The authors thank Anne Lacherez for realizing the stimuli, Julie Brisson for providing both the experimental device and her know-how to be able to use it, and Celine Quint for the proofreading.

## Author Contributions

**Conceptualization:** Pauline Rasset, Benoît Montalan, Jessica Mange.

**Data curation:** Pauline Rasset.

**Formal analysis:** Pauline Rasset.

**Funding acquisition:** Pauline Rasset, Benoît Montalan, Jessica Mange.

**Investigation:** Pauline Rasset.

**Methodology:** Pauline Rasset.

**Project administration:** Pauline Rasset, Benoît Montalan, Jessica Mange.

**Resources:** Pauline Rasset, Jessica Mange.

**Software:** Pauline Rasset.

**Supervision:** Benoît Montalan, Jessica Mange.

**Validation:** Pauline Rasset, Benoît Montalan, Jessica Mange.

**Visualization:** Pauline Rasset.

**Writing – original draft:** Pauline Rasset.

**Writing – review & editing:** Pauline Rasset, Benoît Montalan, Jessica Mange.

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
