## [Decision Letter · Decision Letter 0]

15 Sep 2023

PONE-D-23-17359Only human after all? A pre-registered study on gaze behavior and humanity attributions to people with facial differencePLOS ONE

Dear Dr. Rasset,

Thank you for submitting your manuscript to PLOS ONE. After careful consideration, we feel that it has merit but does not fully meet PLOS ONE’s publication criteria as it currently stands. Therefore, we invite you to submit a revised version of the manuscript that addresses the points raised during the review process. Both reviewers found merit in your work but have provided a number of comments and questions. Please carefully consider a response to each of their points.

We look forward to receiving your revised manuscript.

Kind regards,

Jim Uttley, Ph.D

Academic Editor

PLOS ONE

Journal Requirements:

“This study was supported by the Normandy Region [grant RIN 100%, 2018] to PR, and a grant from the Foundation Gueules Cassées [22 910 €, 2020] to PR. These funders did not play any role in the study design, data collection and analysis, decision to publish, or preparations of the manuscript.”

Reviewers' comments:

Reviewer's Responses to Questions

**Comments to the Author**

1. Is the manuscript technically sound, and do the data support the conclusions?

Reviewer #1: Yes

Reviewer #2: Partly

2. Has the statistical analysis been performed appropriately and rigorously? 

Reviewer #1: I Don't Know

Reviewer #2: Yes

3. Have the authors made all data underlying the findings in their manuscript fully available?

Reviewer #1: Yes

Reviewer #2: Yes

4. Is the manuscript presented in an intelligible fashion and written in standard English?

Reviewer #1: Yes

Reviewer #2: No

5. Review Comments to the Author

Reviewer #1: This is an interesting study of gaze behaviour and dehumanization in perceptions of people with facial difference (FD). It follows up past eye-tracking work documenting reductions in attention to the eyes of people with FD, which are associated with disgust and might account for negative social responses to them. Although that prediction seems well justified, a pilot study reported in the supplementary materials showed no evidence that people with FD were attributed less humanness (or less mind perception or lower scores on the stereotype content dimensions), and the main study shows that they were attributed more 'human nature". Nevertheless, FD faces were gazed at differently from non-FD faces in the predicted way: more attention to the difference and less to the eyes. There are no reliable correlations between gaze behaviour and the humanness attributions. The main pre-registered predictions regarding humanness were therefore not supported.

The research is competently conducted. The eye-tracking work has been analysed carefully and the measurement of the various humanness traits looks adequate, although no evidence is reported for the validity of the French translations of the chosen traits in representing the two kinds of humanness. The authors are surely correct in interpreting the surprising attribution of greater humanness to the FD faces as being due to social desirability. Unlike the gaze behaviour, the trait ratings are slow, deliberate, reflective, and easily made to be sympathetic to people seen as victims of adversity and stigma. The authors are also correct in proposing that more immediate and less bias-prone measures of dehumanization might be used in future work (e.g., implicit measures might be beneficial here) and that perceptions of being dehumanized from the target's perspective should also be examined.

Overall I think this research is useful, even if the results fail to support the credible dehumanization hypothesis. I only have a few suggestions or criticisms.

1. it is not clear in the main study how the HU and HN measures have been constructed. The analysis seems to have been done differently than in the pilot study, where means for HN and non-HN (and HU and non-HU) traits are presented. No means for the non-HN and non-HU traits are reported in the main study, and when the authors write "participants attributed more HN traits to faces with a disfigurement" it is not clear whether they are referring to the three HN traits or to some combination of the HN and the non-HN traits. Were single scales developed for the main study by reverse-scoring the "non" traits or was some other method employed?

2. It is possible that the higher HN ratings for the FD faces was due not (or not only) to socially desirable responding but to the specific traits used to measure HN or non-HN? Two of the HN traits involve sensitivity, and it's possible people perceived to have suffered pain (e.g., from a burn disfigurement) are seen as more sensitive (both emotionally and morally, as pain is often seen as ennobling people). The salience of their inferred suffering might lead to higher ratings on these items and thus relatively high HN scores. Ideally more attention might be paid post hoc to examining which HN and non-HN items showed differences between FD and other faces, as this might clarify the reasons for the surprising finding. In addition, it would have been desirable to have a longer and more desirable dehumanization measure and/or more than one measure.

3. Given that disgust was part of the rationale for expecting non-eye gaze might be associated with dehumanization, why was disgust not measured in the study?

4. It might also be worth noting on the Discussion that most work on trait attribution and dehumanization has focused on perceptions of groups rather than individuals, and when it has used individuals as targets it usually does not provide (humanizing) faces as stimuli. It is therefore not clear whether trait ratings of individual faces are an optimal way to assess dehumanization.

Reviewer #2: Thank you for inviting me to review this manuscript.

The paper describes an eye-tracking study investigating eye gaze towards the face of a person with a visible facial difference. The results replicated previous studies showing enhanced attention to the area of the visible difference but there was no relationship with measures of dehumanisation. The paper is written concisely and is mostly straightforward though the clarity could be improved in some places.

Major points

I’m not convinced that we should ever have expected that a person with a visible facial difference would be regarded as less human. Humanity is an attribution of the person, whereas the experiences of disgust mentioned in the introduction are a response to the visual appearance of the face, not the person. This distinction could be made more clearly. The indirect findings referred to in the Introduction linking dehumanisation with visible facial difference did not amount to a compelling argument.

The Discussion seems determined to pursue the concept of dehumanisation in future studies but perhaps it would be more worthwhile to consider in what ways people with visible facial difference are regarded as different; dehumanisation may not capture the heart of how they are socially perceived.

Minor points

Page 5 – I could not follow the procedure without knowing how many faces, and how many versions of each face, were presented. The paper appears to have been written to a very tight word count, but even so, a minimal level of information should be included to assist the reader in understanding the research. I appreciate the information is in the supplementary materials, but it needs to be also (briefly) in the main paper.

Page 6 – similarly, UH traits and HN traits should be defined, albeit briefly, in the main paper.

I didn’t understand this point, so please rephrase “Two overarching manifestations of dehumanization have been described: dehumanization as a content of social perception (e.g., attribution of nonspecific human characteristics), and dehumanization as a form of perception (7).”

6. PLOS authors have the option to publish the peer review history of their article (what does this mean?). If published, this will include your full peer review and any attached files.

Reviewer #1: No

Reviewer #2: No

---

## [Author Response · Author response to Decision Letter 0]

26 Oct 2023

All responses are included in the "Response to Reviewers" file.

---

## [Editor Report · Decision Letter 1]

13 Nov 2023

PONE-D-23-17359R1Only human after all? A pre-registered study on gaze behavior and humanity attributions to people with facial differencePLOS ONE

Dear Dr. Rasset,

Thank you for your thoughtful responses to the reviewer comments.

The manuscript is very close to be ready for acceptance. However, I think the manuscript would further benefit from including some of the discussion and response you have given to reviewer #1's comment about the lack of inclusion of a disgust measure in the study. I feel this is a valid question and one that other readers are likely to ask, so including your response to the reviewer's comment somewhere in the manuscript (perhaps in the Discussion?) would be useful. I hope this is not too onerous, and I think this addition would ensure the manuscript can be accepted for publication.

We look forward to receiving your revised manuscript.

Kind regards,

Jim Uttley, Ph.D

Academic Editor

PLOS ONE
---

## [Editor Report · Decision Letter 2]

27 Nov 2023

Only human after all? A pre-registered study on gaze behavior and humanity attributions to people with facial difference

PONE-D-23-17359R2

Dear Dr. Rasset,

Thank you for the minor revision to your manuscript in response to my comment. We’re pleased to inform you that your manuscript has been judged scientifically suitable for publication and will be formally accepted for publication once it meets all outstanding technical requirements.

Kind regards,

Jim Uttley, Ph.D

Academic Editor

PLOS ONE
---

## [Editor Report · Acceptance letter]

4 Dec 2023

PONE-D-23-17359R2 

Only human after all? A pre-registered study on gaze behavior and humanity attributions to people with facial difference 

Dear Dr. Rasset:

I'm pleased to inform you that your manuscript has been deemed suitable for publication in PLOS ONE. Congratulations! Your manuscript is now with our production department. 

Kind regards, 

on behalf of

Dr. Jim Uttley 

Academic Editor

PLOS ONE